# Small GTPase—A Key Role in Host Cell for Coronavirus Infection and a Potential Target for Coronavirus Vaccine Adjuvant Discovery

**DOI:** 10.3390/v14092044

**Published:** 2022-09-14

**Authors:** Wei Hou, Sibei Wang, Heqiong Wu, Linli Xue, Bin Wang, Shouyu Wang, Haidong Wang

**Affiliations:** 1College of Veterinary Medicine, Shanxi Agricultural University, Jinzhong 030801, China; 2Single Molecule Nanometry Laboratory (Sinmolab), Nanjing Agricultural University, Nanjing 210095, China; 3OptiX+ Laboratory, Wuxi 214000, China

**Keywords:** small GTPase, coronavirus, infection, adjuvant

## Abstract

Small GTPases are signaling molecules in regulating key cellular processes (e.g., cell differentiation, proliferation, and motility) as well as subcellular events (e.g., vesicle trafficking), making them key participants, especially in a great array of coronavirus infection processes. In this review, we discuss the role of small GTPases in the coronavirus life cycle, especially pre-entry, endocytosis, intracellular traffic, replication, and egress from the host cell. Furthermore, we also suggest the molecules that have potent adjuvant activity by targeting small GTPases. These studies provide deep insights and references to understand the pathogenesis of coronavirus as well as to propose the potential of small GTPases as targets for adjuvant development.

## 1. Introduction

Coronaviruses (CoVs), belonging to the *Coronaviridea family*, *Nidovirales order*, are a group of large-scale and enveloped viruses with positive-sense, single-stranded RNA genomes [1]. The RNA genomes of CoVs range in size from ~27 to 32 kb and encode 14 open reading frames (ORFs). Based on the serology and phylogenetic clustering, CoVs can be classified into four genera: Alphacoronavirus, Betacoronavirus, Gammacoronavirus, and Deltacoronavirus [2] shown in Figure 1. The symptoms and pathological damages caused by CoVs are different. Some CoVs, like human coronavirus NL63 (HCoV-NL63), human coronavirus 229E (HCoV-229E), human coronavirus OC43 (HCoV-OC43), and human coronavirus HKU1 (HCoV-HKU1), can cause human self-limiting common cold-like illnesses. But severe acute respiratory syndrome coronavirus (SARS-CoV) and Middle East respiratory syndrome coronavirus (MERS-CoV) can induce life-threatening diseases and have pandemic potential [3,4]. The current coronavirus disease 2019 (COVID-19) pandemic caused by SARS-CoV-2 has infected over 539.9 million people and led to the deaths of more than 6.32 million individuals as of late June 2022, thus causing a significant threat to worldwide public health [5]. Additionally, some other CoVs, such as porcine epidemic diarrhea virus (PEDV) and transmissible gastroenteritis virus (TGEV), can cause severe diseases in swine and result in severe economic losses [1].

CoVs require multiple signaling molecules in the host cell to complete their life cycle processes. Among them, small GTPases, especially their Rho, Rab, and Arf subfamilies, are widely exploited by CoVs. The Rho subfamily has the function of controlling actin turnover as well as coordinating cell shape and motility, and is mainly co-opted during the CoV entry [6,7]. The Rab subfamily, which is involved in endocytic vesicle trafficking and vesicle fusion, has been enlisted during the intracellular traffic, replication, and egress of CoVs [7,8,9]. The Arf subfamily, which is involved in a broad spectrum of physiological roles, such as the organization of the cytoskeleton, the sorting of vesicle cargo, the recruitment of vesicle coat proteins, and the vesicle budding in the secretory system, is mainly involved in the late stages of CoV infection, such as viral replication and egress [10,11]. Therefore, small GTPases play a role in the host cell for CoV infection. Besides, these small GTPases and their regulators are important for adjuvant developments. Many molecules, such as the regulators of small GTPases, have been discovered with potential adjuvant properties by targeting the activity of small GTPases. These molecules can stimulate the adaptive immunity and production of antibodies with powerful and long-lasting characteristics [12,13,14], which are required for CoV vaccine adjuvant development. Thus, considering the diverse roles of small GTPases in CoV infection, small GTPases are potential targets for CoV vaccine adjuvant discovery.

In this review, the small GTPase roles in the life cycle of CoVs are summarized, and the current understanding of the interaction of small GTPases with CoVs is also presented. Meanwhile, the molecules targeting small GTPases with adjuvant activity are listed, and the potential of small GTPases as targets for adjuvant development is discussed. This review is beneficial for understanding the pathogenesis of CoVs as well as paves the way toward the small GTPases as targets for adjuvant development.

## 2. Overview of the Regulation and Function of Small GTPases

Small GTPases, belonging to the Ras superfamily, are monomeric guanine nucleotide-binding proteins of low molecular weight (21 to 30 kDa) [15]. These small GTPases participate in signaling cascades that control a wide range of cell responses, such as proliferation, differentiation, and motility [16,17].

The small GTPases are molecular switches that undergo a cycle switch between an active GTP-bound form and an inactive GDP-bound state. Three sets of proteins can regulate these GTP-GDP switches. Guanine nucleotide exchange factors (GEFs) catalyze the activating exchange of GDP for GTP, GTPase activating proteins (GAPs) stimulate the intrinsic GTPase activity to inactivate the switch, and guanine nucleotide-dissociation inhibitors (GDIs) block spontaneous activation [16] (Figure 2). By hydrolyzing GTP, these GTPases act as molecular switches that interact with effector proteins and control the activity and function of a variety of specific targets, including enzymes, scaffolds, and accessory proteins, which are involved in a diverse array of cellular events [16].

According to the sequence homology and the physiological functions, small GTPases, with more than 150 members, are divided into five major subfamilies: Ras, Ran, Rho, Rab, and Arf [18]. Members of these subfamilies share common structural features, which are four guanine nucleotide-binding domains and one effector-binding domain, but have different functions (Figure 2). The Ras subfamily includes 36 different members and is responsible for the activation of intracellular signaling networks involved in enhancing cellular proliferation, adhesion, migration, as well as cell survival [19]. The Ran subfamily consists of only one protein, Ran, which controls molecular export and import from the nucleus to the cytoplasm. The Rho subfamily, comprised of 22 members, is involved in reorganizing the actin cytoskeleton and in coordinating cell shape and movement [20]. The best-characterized members of the Rho GTPases are Rac1, RhoA, and Cdc42 in mammalian cells. Rac1 activation regulates the formation of lamellipodia or membrane ruffles. RhoA primarily promotes the formation of actin stress fibers and the assembly of focal adhesion, whereas Cdc42 induces the formation of protruded filopodia. The Rab subfamily consists of approximately 70 members whose main functions are to control trafficking, docking, budding, and fusion of specific vesicles [21]. Each Rab protein associates with an organelle and specifies a trafficking step along endocytic, exocytic, and recycling pathways [22,23]. The Arf subfamily, with 30 members, controls cellular processes such as bidirectional membrane trafficking (secretion and endocytosis), lipid metabolism, motility, division, apoptosis, and gene transcription [24]. The Arf GTPases are localized in the plasma membrane, endosomes, lipid droplets, mitochondria, and lysosomes and are well recognized for their roles in the recruitment of coat proteins/complexes and initiation of vesicle formation in membrane trafficking, particularly at the Golgi [20,24].

Given the importance of small GTPases in a variety of cellular processes, numerous viruses have evolved diverse interactions with small GTPases and manipulated them for their benefit [25,26,27,28]. CoVs, like many other viruses, employ a similar strategy. In the following, we will focus on the function of small GTPases that have been reported during diverse lifecycles of CoVs.

## 3. Overview of the Role of Small GTPases in Host Cell for CoV Infection

Currently, numerous studies on the CoV life cycle have been reported to explain CoV outbreaks [29]. Briefly, CoV initiates infection by binding to cell receptor(s) before entering the host cell and releasing the genome into the cytoplasm. Shortly after that, the viral membrane can either fuse directly with the host cell, or viruses are internalized via endocytosis [8]. Subsequently, CoV releases genomic RNA into the cytosol to initiate transcription and translation, capsid maturation, and envelopment, and ultimately virus egress from the host cell [29]. During CoV infection, the small GTPases, especially Rho, Rab, and Arf GTPases, are involved in the diverse processes of the CoV life cycle (Figure 3).

### 3.1. Roles of Rho GTPases in CoV Infection

As the powerful signaling molecules in cells, Rho GTPases are found in all eukaryotic organisms and regulate cell polarity and motility through their effects on the cytoskeleton, membrane trafficking, and cell adhesion [30]. Increasing evidence shows that many viruses evolve diverse interactions with Rho GTPase signaling and manipulate them for their own benefit. Especially in the case of CoV infection, Rho GTPases participate in the processes of pre-entry and endocytosis.

#### 3.1.1. Pre-Entry

The actin filaments, the most abundant polymers in a large number of cells, can construct finger-like protrusions, such as filopodia, which are important for virus land [31]. During pre-entry, CoVs induce the rearrangement of the actin cytoskeleton by activating Rho GTPases, which allows them to surf along the cell surface, ultimately facilitating their entry (Figure 4). A study using porcine hemagglutinating encephalomyelitis virus (PHEV) labeled with the lipophilic fluorescent dye DiD discovered that at the first 10 min post-infection, the virus reached the actin-rich protrusion, and subsequently (at ~30 min post-infection), the virus surfed along the filopodia via actin rearrangement for entry by activating Rac1 and Cdc 42 signaling [6,7]. Similar results were also found during PEDV and TGEV entries. After the viruses bound to IPEC-J2 cells, they were found to move on the cells along filopodia formed by microfilaments, which then gathered around the viruses for viral internalization [32].

#### 3.1.2. Endocytosis

After migrating along the cell surface to a specific receptor(s), the virus-receptor interactions activate signaling cascades that can guide the virus into cells through the clathrin-dependent and/or clathrin-independent endocytosis pathways. The clathrin-independent endocytosis pathways cover a diversity of internalization routes, such as caveolin-dependent endocytosis (CDE), micropinocytosis, and the clathrin-independent carrier/glycosylphosphatidylinositol-anchored protein-enriched endosomal compartments (CLIC/GEEC) pathway. The above types are all hijacked by CoVs and frequently involve Rho GTPase signaling (Figure 4).

Clathrin-mediated endocytosis (CME), which transports a large number of different cargoes from the plasma membrane into the cell, plays a key role in maintaining cell membrane homeostasis and regulating intracellular signaling [33]. As a key regulator of actin dynamics, Rho GTPases affect not only the formation of clathrin-coated vesicles but also the subsequent movement of vesicles [25,34]. A study using siRNA techniques elucidated that mouse hepatitis coronavirus (MHV) entered cells via actin cytoskeleton-dependent CME and was regulated by Rac1 and Cdc42 signaling [35,36,37] (Figure 4). Other CoVs, including MERS-CoV [38], PDCoV [39], HCoV-NL63 [40], SARS-CoV [41] and SARS-CoV-2 [42], have been reported to enter cells also via CME (Figure 4). The interaction between the HCoV-NL63 and the ACE2 molecule could trigger the recruitment of clathrin, followed by clathrin-mediated, dynamin-dependent endocytosis, which relies on actin remodeling regulated by Rho GTPases [40,43]. Furthermore, PHEV invaded N2a cells via CME, which may be dependent on actin assembly as well, as actin kinetics and cofilin activity affected by the integrin α5β1-FAK-Rac1/Cdc42-PAK-LIMK-cofilin pathway contribute to PHEV invasion [6,7] (Figure 4). The same endocytosis pathway has been described during the entry of TGEV and PEDV [8,44] (Figure 4). During their internalization, the epidermal growth factor receptor (EGFR) was activated. The interaction of the TGEV spike protein with the EGFR activated the downstream phosphoinositide-3 kinase (PI3K), which then caused the cofilin phosphorylation and F-actin polymerization via Rac1/Cdc42 GTPases [45]. This study is consistent with the reports that when EGFR in the lipid rafts is stimulated, the endocytosis of membrane microdomains can occur through signaling cascades in clathrin-dependent and/or clathrin-independent mechanisms [46,47].

Caveolae-mediated endocytosis (CavME) is one of the clathrin-independent endocytic pathways involving caveolae, which are 50- to 100-nm bulb-shaped invaginations [48,49]. This pathway has unique signaling machinery and is involved in the internalization of some CoVs (Figure 4), such as HCoV-229E [36], HCoV-OC43 [50], porcine deltacoronavirus (PDCoV) [51], and PEDV [8]. During virus internalization via CavME, Rho GTPases may play a role in the budding of caveolae. Upon stimulation, RhoA and Rac1 can be recruited to caveolae, and RhoA has been shown to interact directly with caveolae [52,53]. Moreover, the Rho GTPase-regulated actin stress fibers can affect the linear distribution of many types of caveolae in the plasma membrane [54]. Furthermore, Rho GTPases have been proposed to regulate caveolae formation [55]. Interestingly, CavME, in turn, can also regulate the activity and localization of Rho GTPases [56], as well as induce the depolymerization and mobilization of Rho GTPase-dependent actin rearrangement [57].

Macropinocytosis is a transient, actin-dependent cellular process that leads to the internalization of fluid and membrane into large vacuoles and is widely used by viruses, including CoVs [58,59] (Figure 4). In this pathway, the interaction between the virus and the cell membrane triggers the intracellular signals that are necessary to induce the membrane blebbing and ruffles as well as the formation of macropinosomes [59]. The intracellular signals include multiple small GTPases, such as Rac1 and Cdc42, which are responsible for triggering the membrane ruffles of macropinocytosis by activating the effectors of actin polymerization and stability, as well as the effectors of myosin-dependent contraction [60,61,62,63]. It was reported that PDCoV and PEDV entered cells both through the macropinocytosis pathway [39,58]. Moreover, during internalization, the tight junction protein occluding, as the entry factor of PEDV, was also internalized with PEDV through the macropinocytosis pathway, revealing a new mechanism of PEDV infection [58]. In addition, macropinocytic uptake has also been suggested for SARS-CoV, as SARS-CoV can induce membrane ruffles and share some signaling molecules with macropinosome formation, such as PI3K, vimentin, Abl, and Ras, during entry [64]. More recently, the function of macropinocytosis in SARS-CoV-2 entry has also been investigated. It was described that the macropinocytosis inhibitor EIPA substantially decreased the concentration of viral RNA in the supernatant of SARS-CoV-2-infected Vero E6 cells [65].

The other endocytic route that CoVs enter is the CLIC/GEEC pathway, which is regulated by small GTPases like Cdc42 [66] (Figure 4). A report showed that the receptor-binding domain (RBD) of SARS-CoV-2 spike protein was internalized in human gastric-adenocarcinoma (AGS) cells via the pH-dependent CLIC/GEEC endocytic pathway, which may propose a new strategy to target SARS-CoV-2 entry [67].

### 3.2. Roles of Rab GTPases in CoV Infection

Rab GTPases, the largest family of small GTPases, are key regulators of intracellular itineraries, particularly in endocytic vesicle trafficking and vesicle fusion [22]. This mechanism has been subtly enlisted by viruses [23]. After being endocytosed, viral particles are sequestered in endocytic organelles and transported to designated locations until the appropriate conditions are met for viral-genome release, during which the associated Rab proteins are recruited.

#### 3.2.1. Intracellular Trafficking

After internalization, many viruses are delivered to suitable endosomes and follow the intracellular pathway of the endosomal/lysosomal system. A study using infectious bronchitis virus (IBV) labeled with octadecyl rhodamine (R18) revealed that the virus moved along with the classical endosome/lysosome track, in which the activated Rab5 and Rab7 were required [68]. Similar results were also observed in PEDV-infected Vero and IPEC-J2 cells. PEDV colocalized with EEA1 (Rab5), Rab7, and LAMP1 after 30 min, 40 min, and 50 min of endocytosis, respectively. These results revealed that PEDV was transported by specific endosomes and through the early endosome-late endosome-lysosomal pathway after endocytosis [8,69]. In addition, the functional impact of RNAi (RNA interference) mediated gene silencing revealed that the endocytosis-associated proteins EEA1, Rab5, Rab7A, and Rab7B were important for MHV infection [70]. Similarly, the requirement of Rab5 and Rab7, but not Rab11, to transport PDCoV particles after endocytosis was also found by using RNAi and overexpression of the dominant negative mutant of Rab proteins [39]. Furthermore, in a recent study, SARS-CoV-2 was reported to undergo rapid, clathrin-mediated endocytosis in infected cells, and its spike protein could colocalize with Rab5 after 25 min of internalization [42]. Therefore, these studies have revealed that Rab GTPases are required for the intracellular trafficking of CoVs following endocytosis.

As mentioned above, Rab GTPases are powerful tools for discriminating between pathways leading to different intracellular locations. Strikingly, although most CoVs undergo intracellular trafficking via the Rab GTPase-dependent endosome-lysosome intracellular trafficking pathway, the membrane fusion sites of CoVs are quite different [70]. The fusion site for MHV and feline infectious peritonitis virus (FIPV) is in the lysosome (Rab7/LAMP1-positive compartments), whereas MERS-CoV and PHEV occur in the early endosome [7], and PEDV primarily occurs in the late endosome (Rab7-positive compartment) [8] (Figure 5). Given the growing number of proteases that have been shown to cleave the CoV spike proteins [71], CoVs evolve to fuse in different organelles, probably related to the proteolytic enzymes available in CoV target tissues and cells in vivo.

#### 3.2.2. Replication

As the acidic environment of endosomes can facilitate the infiltration of the incoming viruses into the host cytoplasm, thus, Rab GTPases, especially Rab5 and Rab7, are used for the productive infection of CoVs (Figure 5). In PHEV-infected mouse neuroblastoma (Neuro-2α) cells, Rab GTPases are not only involved in the trafficking of internalized PHEV but also play a crucial role in viral proliferation. By using the DiD-labeled PHEV, the colocalization between PHEV and Rab5 or Rab7 was found. Furthermore, a GTPase activation assay in this study suggested that the high-GTPase-activity isoform Rab5 could facilitate PHEV RNA replication and proliferation, while the dominant negative isoform Rab5 significantly inhibited the productive infection of PHEV [7]. Likewise, a study reported that silencing Rab5 and Rab7 could notably reduce viral RNA copy numbers and N protein expression levels of PDCoV via RT-qPCR and Western blot analysis [39]. Moreover, it was also reported that PDCoV and its entry cofactor aminopeptidase N (pAPN) colocalized with the endocytotic markers Rab5, Rab7, and LAMP1, suggesting that pAPN mediates PDCoV entry by an endocytotic pathway. More importantly, it was emphasized that regardless of receptor usage, only PDCoV entry via an endocytosis route ultimately leads to efficient replication [72]. These studies highlight the significance of Rab GTPases in CoV replication.

#### 3.2.3. Egress

After the viral genome is transported to specific sites in the cytoplasm for replication and assembly, an increasing number of viruses turn out to exploit the endocytic recycling apparatus defined by Rab11 to egress from their host cells [9,26,27,28,73,74,75]. CoVs, like many other enveloped viruses that bud either intracellularly or at the cell surface, have recently been reported to employ Rab11 to gain exit from their host cell [9] (Figure 5). IBV, as a well-established model virus, is used to investigate the pathway of CoV egress from epithelial Vero cells. The result showed that IBV bypassing the Golgi stacks was based on a direct connection between the intermediate compartment (IC) and the endocytic recycling defined by Rab1 and Rab11, respectively. In this study, the endocytic recycling system provided the carriers for the final delivery of the virus for exocytosis. Interestingly, using IBV as a model virus to investigate the egress pathway of CoV showed that the M protein of IBV in the IC elements was colocalized with Rab11, while negligible overlap with LAMP-1 was observed, indicating that IBV (γ-CoV) egress does not occur via late endosomes or lysosomes, which is different from β-CoVs (SARS-CoV-2 and MHV) using Rab7 GTPases dependent-lysosomes for egress (the detail will be described in the roles of Arf GTPase in CoV infection) [10]. Strikingly, even CoVs from the same genus, such as SARS-CoV and SARS-CoV-2, which both belong to β-CoVs, also have different exit strategies due to differences in their protein sequences [76]. Thus, it is not surprising that CoVs of different genera egress in various ways.

### 3.3. Roles of Arf GTPases in CoV Infection

The ADP-ribosylation factor (ARF) small GTPases, including Arl (Arf-like) GTPases, are best known for their roles in membrane trafficking and vesicle sorting [77]. Similar to other GTPases, Arf GTPases act as molecular switches by shuttling between their active GTP-bound and inactive GDP-bound conformations, which are regulated by GTPase-activating proteins (GAPs) and guanine nucleotide exchange factors (GEFs). Besides Rab GTPases, Arf GTPases are also involved in the late stages of CoV infection as viral replication and egress (Figure 6).

In MHV infection, the two Arf GTPases, Sar1 and ARF1, are both involved in MHV replication. There are two major steps in the anterograde protein secretion route that are linked to MHV replication complex (RC) formation and/or RNA replication [11]. First, the transport of proteins out of the endoplasmic reticulum (ER) requires ER exit site formation controlled by Sar1. Blocking this early step by expressing a dominant mutant of Sar1 blocks MHV replication profoundly [78]. Next, ER exit sites develop into or form de novo, vesicular-tubular clusters (also called ERGIC), for which ARF1 is required. This step, which is also involved in MHV RC formation, can be blocked by expressing a dominant-negative mutant of ARF1 or by down-regulating ARF1 using siRNA [11].

Another Arf GTPase, Arl8b, is a small Arf-like Ras family GTPase that localizes to the late endosomes/lysosomes and regulates their movement to the plasma membrane and, ultimately, their exocytosis. It was reported that Arl8b was not involved in MHV replication, but was for MHV egress [10]. A recent study using imaging techniques and virus-specific reporters revealed that β-coronaviruses, including SARS-CoV-2 and MHV, used lysosomal trafficking for egress, which was regulated by Arl8b and Rab7 GTPases, rather than the more common biosynthetic secretory pathways used by other enveloped viruses [79,80]. In this research, the progeny viruses were released from cells via lysosomal exocytosis. It is known that the subcellular localization of lysosomes is determined by the balance between Rab7 and Arl8b [81], but it is unclear how viruses enter lysosomes and which viral proteins or related host proteins are involved in the process of virus egress. It was demonstrated that ORF3a of SARS-CoV-2, but not SARS-CoV, could promote lysosomal targeting of the BORC-Arl8b complex and exocytosis-related SNARE protein to promote lysosomal exocytosis. Moreover, it was also found that the reason for the differential function of SARS-CoV-2 and SARS-CoV ORF3a in lysosomal exocytosis was due to the residues 171 and 193 of ORF3a [76], and this difference is critical for us to understand the mechanism by which endows SARS-CoV-2 with much higher infectivity and pathogenicity than SARS-CoV. Furthermore, as Arl8 GTPase can bind kinesin1 through its effector SKIP (SifA and kinesin-interacting protein) to promote lysosomes towards the cell periphery by moving along the microtubule with the plus end-directed transport [82], CoVs might undergo the Arl8b GTPase-mediated anterograde movement along the microtubule toward the cell membrane, and this possibility still needs further experiments to confirm.

The above highlights multiple functions of small GTPases during CoV entry, intracellular trafficking, replication, and exit from the host cell, as summarized in Table 1. The interactions of small GTPases and CoVs not only reveal how these signaling molecules engage in the viral replication cycle but also demonstrate knowledge of the processes in which they are naturally involved. However, there are a lot of GTPases belonging to these three subfamilies, and even many small GTPases involved in the same cellular functions, such as Rab5, Rab7, Rab14, and Rab36, all involved in the intracellular transport of cargoes [22,83]. Therefore, the crosstalk between these various proteins and related signaling axes will further complicate this field of study. In addition, while the well-studied members of the Rho, Rab, and Arf GTPases are very important viral targeted small GTPases, other currently un(der)studied small GTPases could also be important for viral infection. Furthermore, the timing, duration, and subcellular localization of specific small GTPases are also important for their functions [25]. Thus, the specific impact of the different small GTPases in virus infection merits further attention in the future. For many viruses, including CoVs, the involvement of small GTPase signaling during infection is still largely based on inhibitor studies or some static research methods. Therefore, direct and dynamic research methods will be more welcomed and encouraged to precisely and in-depth reveal the infection mechanism of CoVs as well as other viruses.

## 4. The Potential of Small GTPases as Adjuvant Targets in the Sites Related to CoV Infection

As a result of the emerging coronavirus pandemic, there is an urgent need to develop effective and safe vaccines that can be rapidly deployed globally [84,85]. Adjuvants are key components of both subunit and some inactivated vaccines because they induce specific immune responses that are stronger and longer-lasting [86,87]. However, there are only a few adjuvants that have received Food and Drug Administration (FDA) approval and are reported in CoV vaccines, such as aluminum salts, MF-59, and AS03 [88,89,90]. Aluminum salts, whose adjuvant properties were discovered about 90 years ago, are the most widely used because of their wide-spectrum ability to strengthen immune responses and their excellent track record of safety [91,92,93]. However, a low level of Th1 CD4^+^ T cell and cytotoxic CD8^+^ T cell immunological responses, which are characteristic of alum-adjuvanted vaccinations, was observed when alum was employed as an adjuvant in CoV vaccines [90]. MF59 and AS03, in contrast to alum, can elicit more balanced immunity in CoV vaccines, potentially by increasing antigen absorption, attracting immune cells, and encouraging the migration of activated antigen-presenting cells. However, AS03 as an adjuvant has been reported to have some safety concerns, as it is associated with narcolepsy in some countries [94]. Additionally, due to various antigen types, MF59 can result in distinct cell-mediated immunity when used as an adjuvant in CoV vaccines. For example, when formulated with the MERS-CoV S protein, MF59 enhanced both effective CD4^+^ and CD8^+^ T cell responses; whereas when combined with inactivated SARS-CoV, MF59 only induced CD4^+^ T cell but not CD8^+^ T cell responses [95,96]. Furthermore, it was also reported that the serum derived from mice immunized with MERS S at various doses in the presence of MF59 did not significantly differ in the neutralizing activity due to the dose-sparing effect of MF59 when it was formulated with MERS S protein [96]. Thus, it is still urgent to develop adjuvants with unique safety and efficacy profiles for the CoV vaccine.

Small GTPase, a potent signaling molecule in cells, is not only involved in many CoV infection processes but also may be a potential target of adjuvant for the CoV vaccine. Although few CoV adjuvants targeting small GTPases have been discovered, compounds with adjuvant capabilities targeting small GTPases are being explored in several aspects related to SARS-CoV-2 infection.

The Rho GTPase activators, cytotoxic necrosis factor 1 (CNF1), and dermal necrosis toxin (DNT) have been reported to have adjuvant properties in the mucosa, the primary site of SARS-CoV-2 infection in the upper respiratory tract. CNF1 is a 114 kDa protein that belongs to a family of bacterial toxins. Once inside the cytosol, CNF1 catalyzes a reversible activation of the Rho GTPases of Rac and Cdc42 by inducing a counterintuitive mechanism that activates the Rho protein by deamidation [97,98]. A study reported that the endothelial cells exposed to high doses of CNF1 could result in a large array of immunomodulators, like IL-8, MCP-1, and MIP-3α, and this was dependent on the function of CNF1 in Rac or Cdc42 activation [99]. Moreover, in the model of intranasal vaccination against tetanus toxin, CNF1 was identified as an effective immunoadjuvant, as it could elicit a specific and durable anti-tetanus toxin response in immunized mice [14]. In addition, a study in mice co-fed with the toxin and the soluble protein antigen ovalbumin (OVA) reported that CNF1 could elicit adjuvanticity anti-OVA responses in mucosal [12]. A similar result was also found in DNT. It was reported that the catalytic domain of DNT, which deamidates and transglutaminases Rho proteins, could stimulate the adaptive immunity and production of antibodies to orally co-administered ovalbumin [12]. These studies suggest that the Rho GTPases are major mediators of the immune responses, and manipulation of Rho GTPase activity can propose a new adjuvant strategy to modulate the mucosal immune responses.

In addition, a recent study has shown that SARS-CoV-2 specific adaptive immune responses are associated with milder disease, and CD4^+^ and CD8^+^ T cell synergistic responses play a synergistic role in protective immunity against COVID-19 [100]. Dendritic cells (DCs), a master for inducting and regulating immune responses involving both CD4^+^ and CD8^+^ T cells, are widely distributed in the respiratory tract and act as important sentinels [101]. Leptin is an adipocyte-derived hormone/cytokine that has an important role in the immune responses of DCs. Studies showed that leptin could promote cofilin activation and cytoskeleton rearrangement by activating Rac1 and triggering Vav phosphorylation, thus improving the migration performance of immature DCs, which functionally upregulated L-12p70 production on CD40 stimulation in immature DCs and increased their capacity to activate autologous CD8^+^ T cells [13]. Therefore, leptin can represent an optimal candidate adjuvant for SARS-CoV-2 vaccination.

Furthermore, lipophilic statins (e.g., simvastatin) and lipophilic bisphosphonates are also potent vaccine adjuvants and have been demonstrated in mice and cynomolgus monkeys. These adjuvants target the small GTPases by inhibiting the geranylgeranylation of small GTPases (e.g., Rab5), resulting in arrested endosomal maturation, prolonged antigen retention, enhanced antigen presentation, and T cell activation [88]. Moreover, statins are low-cost, extensively tested, and well-tolerated drugs that are also supported as adjunctive therapy in the clinical management of COVID-19 patients [102]. Furthermore, two nonsteroidal anti-inflammatory drugs, R-naproxen and R-ketorolac, have been reported as potential adjuvants in cancer therapy, with similar hyperinflammatory conditions to COVID-19 [103,104]. They can block the activation of Rac and Cdc42 GTPases in response to growth factor stimulus, as well as downstream cellular responses that depend on these activated GTPases, like cell proliferation, migration, and adhesion [104]. Given the function of Rho GTPases in CoV infection, R-naproxen and R-ketorolac have great potential as adjuvants for CoVs. But it still needs further verification.

Significantly, similar to aluminum salts, MF-59, and AS03, the above compounds targeting small GTPases also have good adjuvant capabilities (Table 2), and in some aspects, even better than these three adjuvants, such as CNF1 is more potent than alum in inducing mucosal IgA antibody responses [12,94]. Therefore, with the diverse roles of small GTPases in CoV infection, targeting small GTPases for the development of CoV vaccine adjuvants will be a new and promising research area. 

The adjuvant is a critical component of vaccine development, but its paucity of mechanism research on the targets limits its further application [88]. As small GTPases with a large number of them involved in the same cellular functions [22], their role depends on the timing, duration, and subcellular location of the signal [25]. Therefore, the role of specific small GTPases at the subcellular localization and signaling pathway level in the CoV life cycle is a key point. It will be helpful to find the small GTPase target. In addition, to achieve the small GTPases as adjuvant targets, two additional problems need to be solved. One is to select the compounds or cellular factors that can mediate the function of the target small GTPases. Another is to screen for compounds or cellular factors with adjuvant activity and long-term safety, such as Rho GTPase activators, CNF1, and DNT having adjuvant properties [12].

## 5. Conclusions

As obligate cellular parasites with limited genomic capacities, CoVs have evolved to effectively utilize intracellular factors to facilitate their life cycle, including entry, intracellular trafficking, replication, and egress. Here, we summarize and highlight the roles of small GTPases in the life cycle of CoVs. Additionally, we discuss the molecules that target small GTP and have potent adjuvant activity, and we suggest small GTPase as an adjuvant target for CoV vaccine development. This review is beneficial to understand the function of small GTPases in CoV infection and may help to find new adjuvants for CoV vaccines.

## Figures and Tables

**Figure 1 viruses-14-02044-f001:**
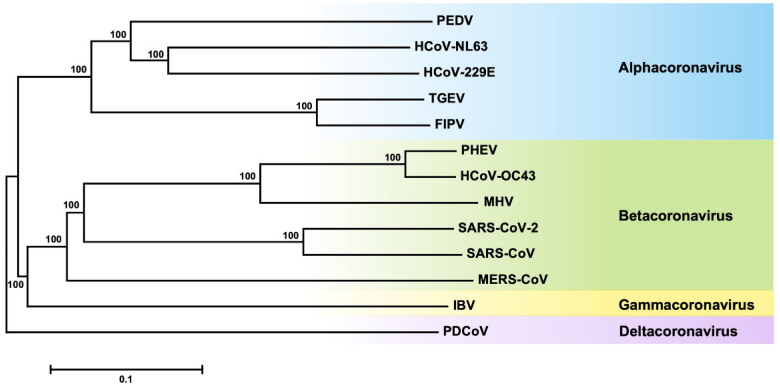
Phylogenetic tree of CoVs. The tree was constructed using the Neighbor-Joining method via 1000 bootstrap replicates with the MEGA 7.0 program. The scale bar represents the evolutionary distance in substitutions per site.

**Figure 2 viruses-14-02044-f002:**
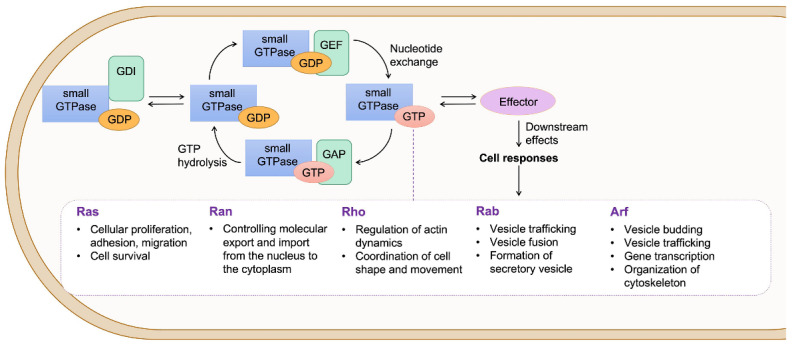
The activation/inactivation cycle and function of small GTPases. GAP binding induces the GTP hydrolysis and inactivation of small GTPases. GDIs sequester small GTPases and mediate intermembrane transport by forming soluble complexes. GEF-mediated nucleotide exchange activates GDP-bound small GTPases. In their GTP-bound form, they interact with effector proteins to trigger downstream signaling events. The most representative functions of small GTPases are listed in the box.

**Figure 3 viruses-14-02044-f003:**
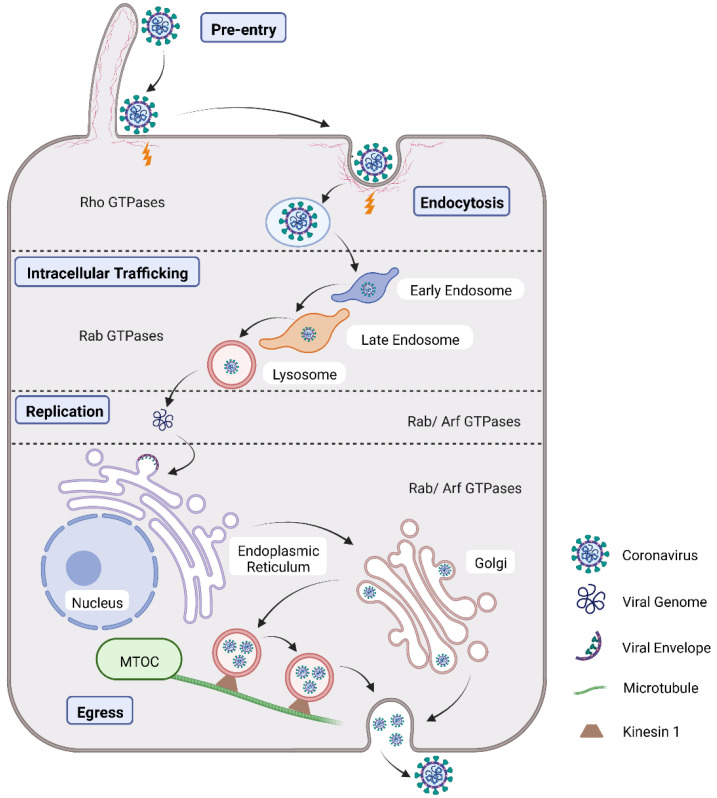
CoV life cycle scheme. Virion surfs along filopodia in pre-entry, then internalization via endocytosis, traffic in the vesicles, genome delivery and replication in the cytoplasm, and ultimately egress from the host cell with small GTPases (Rho GTPases, Rab GTPases, and Arf GTPases) involved.

**Figure 4 viruses-14-02044-f004:**
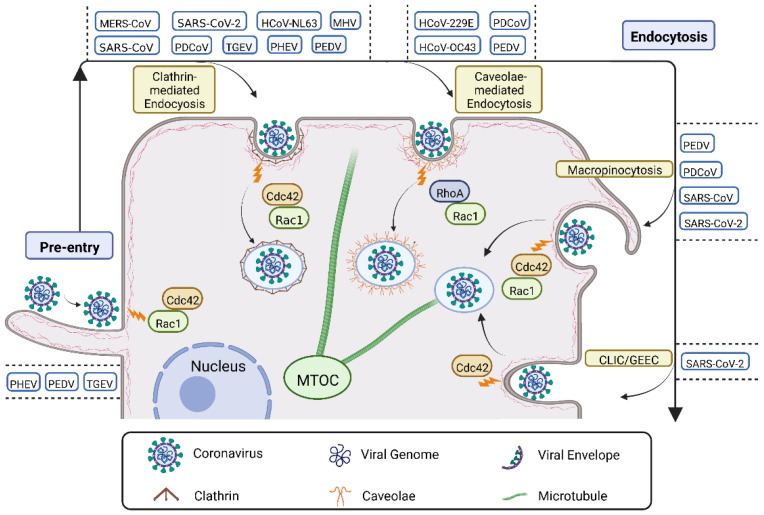
An overview of the roles of Rho GTPases in CoV infection. During pre-entry, CoVs (PEDV, TGEV, and PHEV) attach to the plasma membrane, and subsequently, land on the cell surface along the filopodia to locate specific receptor(s), which are Rac1 and Cdc42 dependent. The interaction between the specific virus-receptors can activate signaling cascades, guiding the virus internalization through one or more of the following pathways: clathrin-mediated endocytosis for MERS-CoV, PDCoV, HCoV-NL63, PHEV, MHV, SARS-CoV, SARS-CoV-2, TGEV, and PEDV infection with Cdc42 and Rac1 dependent; caveolae-mediated endocytosis for HCoV-229E, HCoV-OC43, PDCoV, and PEDV infection with RhoA and Rac1 dependent; macropinocytosis for PEDV, PDCoV, SARS-CoV, and SARS-CoV-2 infection with Rac1 and Cdc42 dependent; and CLIC/GEEC for SARS-CoV-2 infection with Cdc42 dependent. *Abbreviations*—MTOC: microtubule organizing center; CLIC/GEEC: clathrin-independent carrier/glycosylphosphatidylinositol-anchored protein-enriched endosomal compartments; PHEV: hemagglutinating encephalomyelitis virus; PEDV: porcine epidemic diarrhea virus; TGEV: transmissible gastroenteritis virus; MERS-CoV: Middle East respiratory syndrome coronavirus; SARS-CoV: severe acute respiratory syndrome coronavirus; PDCoV: porcine deltacoronavirus; MHV: mouse hepatitis coronavirus; HCoV-NL63: human coronavirus NL63; HCoV-229E: human coronavirus 229E; HCoV-OC43: human coronavirus OC43.

**Figure 5 viruses-14-02044-f005:**
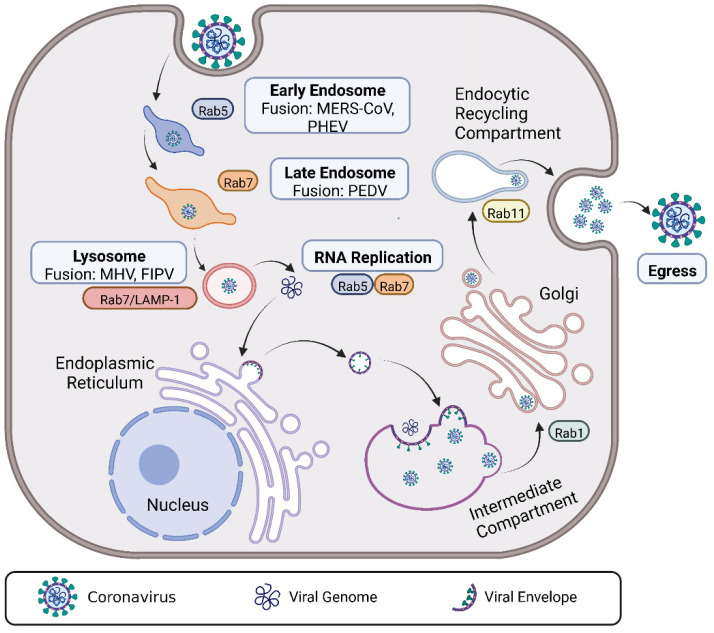
An overview of the roles of Rab GTPases in CoV infection. After internalization, CoVs are delivered to suitable endosomes and follow the intracellular pathway of the endosomal/lysosomal system with diverse Rab GTPases involved. Subsequently, CoVs fuse at the designated locations, such as MERS-CoV and PHEV fusion, which both occur in the early endosome; PEDV fusion mainly occurs in the late endosome; MHV and FIPV fusions occur in the lysosome. In addition, Rab GTPases also play an important role in CoV replication and egress. For example, the replication of PHEV and PDCoV genomes needs the help of Rab5 and Rab7, and IBV, as a well-established model virus used to investigate the pathway of CoV, egress from host cells needs to employ Rab1 and Rab11. *Abbreviations*—MERS-CoV: Middle East respiratory syndrome coronavirus; PHEV: hemagglutinating encephalomyelitis virus; PEDV: porcine epidemic diarrhea virus; MHV: mouse hepatitis coronavirus; PDCoV: porcine deltacoronavirus; FIPV: feline infectious peritonitis virus.

**Figure 6 viruses-14-02044-f006:**
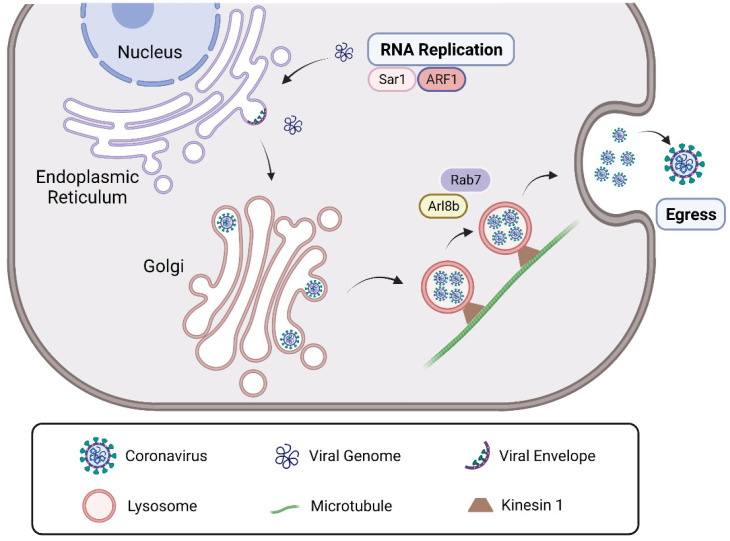
The roles of Arf GTPases in CoV infection. After genome release, MHV will experience RNA replication in the cytoplasm with the help of Sar1 and ARF1 GTPases. After viral replication and assembly, β-coronaviruses such as SARS-CoV-2 and MHV packed by lysosomes may undergo an anterograde movement along microtubules for egress, which is regulated by Arl8b and Rab7 GTPases. *Abbreviations*—MHV: mouse hepatitis coronavirus; SARS-CoV-2: severe acute respiratory syndrome coronavirus 2.

**Table 1 viruses-14-02044-t001:** Summary of the roles of small GTPases in CoV infection.

Phase	Virus (Genera) *	Small GTPase (Family)	Reference
Pre-entry	PHEV (β)	Rac1 and Cdc42 (Rho GTPase)	[6,7]
	PEDV (α)	Rac1 and Cdc42 (Rho GTPase)	[6,7,32]
	TGEV (α)	Rac1 and Cdc42 (Rho GTPase)	[6,7,32]
Endocytosis (CME)	MHV (β)	Rac1 and Cdc42 (Rho GTPase)	[35,36,37]
	PDCoV (δ)	Rac1 and Cdc42 (Rho GTPase)	[39]
	MERS-CoV (β)	Rac1 and Cdc42 (Rho GTPase)	[25,34,38]
	HCoV-ML63 (α)	Rac1 and Cdc42 (Rho GTPase)	[40,43]
	SARS-CoV (β)	Rac1 and Cdc42 (Rho GTPase)	[25,34,41]
	SARS-CoV-2 (β)	Rac1 and Cdc42 (Rho GTPase)	[25,34,42]
	PHEV (β)	Rac1 and Cdc42 (Rho GTPase)	[6,7]
	PEDV (α)	Rac1 and Cdc42 (Rho GTPase)	[8,25,34]
	TGEV (α)	Rac1 and Cdc42 (Rho GTPase)	[44,45]
Endocytosis (CavME)	HCoV-229E (α)	RhoA and Rac1 (Rho GTPase)	[36,52,55]
	HCoV-OC43 (β)	RhoA and Rac1 (Rho GTPase)	[50,52,55]
	PDCoV (δ)	RhoA and Rac1 (Rho GTPase)	[51,52,55]
	PEDV (α)	RhoA and Rac1 (Rho GTPase)	[8,52,55]
Endocytosis (Macropinocytosis)	PDCoV (δ)	Rac1 and Cdc42 (Rho GTPase)	[39]
	PEDV (α)	Rac1 and Cdc42 (Rho GTPase)	[39,58]
	SARS-CoV (β)	Rac1 and Cdc42 (Rho GTPase)	[39,64]
	SARS-CoV-2 (β)	Rac1 and Cdc42 (Rho GTPase)	[39,65]
Endocytosis (CLIC/GEEC)	SARS-CoV-2 (β)	Cdc42 (Rho GTPase)	[66,67]
Intracellular Trafficking	IBV (γ)	Rab5 and Rab7 (Rab GTPase)	[68]
	PEDV (α)	Rab5 and Rab7 (Rab GTPase)	[8,69]
	MHV (β)	Rab5, Rab7A and Rab7B (Rab GTPase)	[70]
	PDCoV (δ)	Rab5 and Rab7 (Rab GTPase)	[39]
	SARS-CoV-2 (β)	Rab5 (Rab GTPase)	[42]
	FIPV (α)	Rab7 (Rab GTPase)	[7,70]
	MERS-CoV (β)	Rab5 and Rab7 (Rab GTPase)	[7,70]
	PHEV (β)	Rab5 and Rab7 (Rab GTPase)	[7]
Replication	PHEV (β)	Rab5 and Rab7 (Rab GTPase)	[7]
	PDCoV (δ)	Rab5 and Rab7 (Rab GTPase)	[72]
	MHV (β)	Sar1 and ARF1 (Arf GTPase)	[11,78]
Egress	IBV (γ)	Rab1 and Rab11 (Rab GTPase)	[9]
	SARS-CoV (β)	Rab7 (Rab GTPase)	[10]
	SARS-CoV-2 (β)	Rab7 (Rab GTPase)/Arl8b (Arf GTPase)	[10,79,80]
	MHV (β)	Rab7 (Rab GTPase)/Arl8b (Arf GTPase)	[10,79,80]

* TGEV: porcine transmissible gastroenteritis virus; PEDV: porcine epidemic diarrhea virus; HCoV-NL63: human coronavirus NL63; HCoV-229E: human coronavirus 229E; HCoV-OC43: human coronavirus OC43; SARS-CoV: severe acute respiratory syndrome coronavirus; MERS-CoV: Middle East respiratory syndrome coronavirus; PHEV: hemagglutinating encephalomyelitis virus; MHV: mouse hepatitis coronavirus; PDCoV: porcine deltacoronavirus; FIPV: feline infectious peritonitis virus; IBV: infectious bronchitis virus.

**Table 2 viruses-14-02044-t002:** Adjuvant molecules targeted small GTPases.

Molecule (Adjuvant)	Targeted Small GTPase	Cell/Animal Type Studied	Reference
CNF1 *	Rac1 and Cdc42	HEp-2 cell; Females BALB/c Mice	[12]
DNT *	Rac1 and Cdc42	Females BALB/c Mice	[12]
Leptin	Rac1	Dendritic cells (DCs)	[13]
Lipophilic statins and bisphosphonates	Rab5	B16-OVA, TC-1 and B16-F10; Mice and Cynomolgus Monkeys	[88]
R-naproxen and R-ketorolac	Rac and Cdc42	Cell-based and preclinical animal studies have been completed (cell and animal types not reported)	[104]

* CNF1: cytotoxic necrosis factor 1; DNT: dermal necrosis toxin.

## Data Availability

Not applicable.

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
