# Peer review of "Small GTPase—A Key Role in Host Cell for Coronavirus Infection and a Potential Target for Coronavirus Vaccine Adjuvant Discovery"

_viruses, 2022, doi:10.3390/v14092044_

Round 1

Reviewer 1 Report

The review paper by Hou et al: Small GTPase-a key role in host cell for coronavirus infection and a potential target for coronavirus vaccine adjuvant discovery, summarizes the current knowledge on the importance of small GTPases in the life cycle of various coronaviruses. The manuscript is nicely illustrated and well presented. The topic is very interesting but extremely complex, which is not adequately presented in this work. Moreover, the importance of small GTPases and their regulators as vaccine adjuvants is a difficult area that could even be written as a separate review. In this review, the authors have chosen to give us only a general overview of the issue. The good side is the broad view of the manipulation of host signaling pathways by coronaviruses. However, a deep and critical account of the problem is lacking. Review of the literature on the role of small GTPases in coronavirus infection deserves a critical analysis of the existing data from the perspective of recent advances in understanding the complexity and redundancy of the regulatory network of small GTPases.

Specific comment:

The authors have described and mentioned many different coronaviruses from four genera that manipulate host small GTPases for the purpose of endocytosis, replication, intracellular trafficking, or egress. It would be helpful to provide a summary table that provides an overview:

- a coronavirus species,

- small GTPase (name and family) used by the specific virus,

- The purpose of the manipulation (internalization, egress...),

- Reference(s).

Major comments:

-          - The authors insufficiently discuss and emphasize the problem of studying the huge number of different small GTPases that label very complex intracellular transport system. The results from the literature should be more extensively analyzed, including the methodology used. Indeed, there are a lot of questions in interpreting the results with small GTPases that act redundantly together with their GAPs and GEFs and are part of a very complex interactome. For example, as the authors mentioned, there are about 70 different Rab GTPases in eukaryotic cells. The large number is not the only problem. Most of them can be expressed in different intracellular localizations, which makes it very difficult to draw a conclusion. For example, Rab11A is a known marker for the endocytic recycling compartment, but can also be found in Golgi and early endosomes. Rab11B is found in ERC, Golgi, TGN, lysosomes and lysosomal exosomes. Rab10 is found in the intermediate EE-ERC compartment but can also be found on Golgi and ER membranes. Although Rab5 and Rab7 are known markers for early endosomes and late endosomes, respectively, they can be visualized at the interface with the endoplasmic reticulum, considering that the ER tubules can wrap around endosomes. Rab5 can also be found on exosomes. Rab7 can be found in autophagosomes. Furthermore, although the best known Rabs, such as Rab4, Rab5A/B/C, Rab7A/B, Rab11A/B, and Rab9, are the most studied, there are a large number of insufficiently studied Rab molecules that are also recognized as very important for intracellular transport, such as Rab8, Rab10, Rab22A, Rab35, Rab14, Rab36... They ensure additional control and provide important information about the transport of the studied molecules/viruses/particles. Unfortunately, they are rarely considered in current studies, especially in virology, which is also a problem that should be discussed.

-          - The  potential of small GTPases as adjuvant targets is very interesting. However, the problems of that approach are also not critically discussed.

Minor comments:

-         - It would be helpful if the general scheme of the CoV life cycle(s) could be presented at the beginning. This would help for further following of the intracellular pathways of the pathogen..

-          Figure 4. Rab11 is shown on Golgi. It could be, but it is generally presented as a marker of ERC, as the authors also discuss in Section 3.2.3. ERC is not even shown in Figure 4.

-      Figure 4: line 235. “MERS-CoV and PHEV fusion, which both occur in lysosome”. Does “lysosome” should be replaced with “early endosome” as presented in Figure 4?

Technical comment:

-         - Perhaps the position of figure 3 and figure 4 could be moved (to the place after they are mentioned)

Reviewer 2 Report

In this study, the authors summarize the roles of small GTPases in the life cycle of CoVs, and discuss the molecules that target small GTP and have potent adjuvant activity. Overall, they give a detailed review of small GTPase roles in CoV infection. However, some statements are confusing, not correct or updated. They also need an English native speaker to polish the language. Here are some comments for further improvement:

Show the full names of the abbreviations in figure 3 legend.

What factors determine the different mechanisms of CoV entry into cells? Such as SARS-CoV2 endocytosis, macropinocytosis, and CLIC/GEEC.

In figure 3, what does it mean in the left side? Only PHEV, PEDV, TGEV move along filopodia in pre-entry?

SARS-CoV-2 also enter the cell by direct membrane fusion and genome release. Any involvement of small GTPase?

It seems not clear about the significance of Rab GTPases in CoV replication. How the colocalization data of CoV and GTPase demonstrate the role of GTPase in replication? It also may affect the endocytosis or trafficking to further affect the viral production or yield. Any direct evidences of GTPase effect on RNA replication? What is the possible mechanism?

In figure 5, how the Sar1 and ARF1 affect the CoV viral RNA replication?

Line 363, “However, there are only three adjuvants that have received Food and Drug Administration (FDA) approval and reported in CoV vaccines: aluminum salts, MF-59, and AS03”. This statement is not correct. Only I know also include CpG, AS04, AS01 etc.

Line 368, “a lack of Th1 CD4+ T cell and cytotoxic CD8+ T cell immunological responses”. “lack” is not suitable although Th1 immune resposnes of alum is not significant compared with Th2.

“it will result in distinct cell-mediated immunity, such as CD8+ T cell responses or not, and even MF59 can produce no significant difference in serum neutralizing activity in antigen immunized mice due to it dose-sparing effect [95,96].” This is confusing.

Line 34, “The current coronavirus disease 2019 (COVID-19) pandemic caused by SARS-CoV-2 has infected over 2.3 million people, and led to the death of more than 160,000 individuals”. Numbers are not most updated.

Lines 16, 49, 204, 233…. It should be “intracellular”? Please check the whole manuscript. I am confused with the usage of “intracellular” and “intercellular” in this manuscript.

Line 57, “These molecules can stimulate the systemic and humoral immune responses”, “systemic” usually include “humoral” and “cellular” responses. What the authors indicate here?

Line 63, replace “demonstrated” with “presented”.

Lines 135 and 146, replace “surf” with “land”. And “surf” may not be suitable in other content.

Line 154, should be “move on”?

Round 2

Reviewer 1 Report

The authors have improved the manuscript according to the comments and it is now better explained. I consider this review suitable to be accepted for publication.

Reviewer 2 Report

All my comments have been addressed.